# Mitochondrial Dysfunction in Pulmonary Hypertension

**DOI:** 10.3390/antiox12020372

**Published:** 2023-02-03

**Authors:** Gusty Rizky Teguh Ryanto, Ratoe Suraya, Tatsuya Nagano

**Affiliations:** 1Laboratory of Clinical Pharmaceutical Science, Kobe Pharmaceutical University, Kobe 658-8558, Japan; 2Division of Respiratory Medicine, Department of Internal Medicine, Kobe University Graduate School of Medicine, Kobe 650-0017, Japan

**Keywords:** pulmonary hypertension, mitochondrial dysfunction, Warburg effect, oxidative stress, pulmonary vascular remodeling

## Abstract

Pulmonary hypertension (PH) is a multi-etiological condition with a similar hemodynamic clinical sign and end result of right heart failure. Although its causes vary, a similar link across all the classifications is the presence of mitochondrial dysfunction. Mitochondria, as the powerhouse of the cells, hold a number of vital roles in maintaining normal cellular homeostasis, including the pulmonary vascular cells. As such, any disturbance in the normal functions of mitochondria could lead to major pathological consequences. The Warburg effect has been established as a major finding in PH conditions, but other mitochondria-related metabolic and oxidative stress factors have also been reported, making important contributions to the progression of pulmonary vascular remodeling that is commonly found in PH pathophysiology. In this review, we will discuss the role of the mitochondria in maintaining a normal vasculature, how it could be altered during pulmonary vascular remodeling, and the therapeutic options available that can treat its dysfunction.

## 1. Introduction

As a condition of varying etiological origins, not many things, aside from the well-known hemodynamic criteria, are common between the many groups of pulmonary hypertension (PH) [1,2]. One the few common denominators found in the molecular changes occurring in PH is the fact that mitochondrial dysfunction, and subsequently oxidative stress, is a major contributor to said changes [3]. Indeed, vascular remodeling commonly found in various forms of PH have been linked to errors in normal mitochondrial function, and the means to combat this problem have been developed over the years [4]. Here, we will review the role of mitochondria in PH pathogenesis from the lens of the cells involved and the process altered. We will provide evidence of mitochondria dysfunction in the clinical setting. We will also further discuss the ways that have been developed to combat dysfunction and the increase in oxidative stress that occurs as a consequence of mitochondria dysfunction, in the hope of spurring further innovation in this specific topic in the future.

## 2. Pulmonary Vascular Remodeling and the Role of Mitochondria

### 2.1. Mitochondria and Vascular Homeostasis

Pulmonary hypertension (PH) is a condition where an elevation in mean pulmonary artery pressure (mPAP) occurs due to varying etiological factors, and the difference in the etiological causes have led the World Health Organization (WHO) to classify it into five different groups [2,5]. The groups include pulmonary arterial hypertension (PAH), PH due to left heart disease, PH due to hypoxia or chronic lung diseases, chronic thromboembolic pulmonary hypertension (CTEPH), and PH due to other causes [2]. Whilst the cause may vary, a common theme is how molecular changes in the pulmonary vasculature occur that can drive structural and eventual hemodynamic changes that can be found clinically [6]. Pulmonary vascular remodeling, as it is widely known today, is now becoming a vital process for researchers to study and for clinicians to target therapeutically, due to the fact that many of the clinical consequences of PH can be linked to this particular process [6]. Alterations to the components of the pulmonary vasculature, namely the endothelial cells (ECs), smooth muscle cells (SMCs), and adjacent mesenchymal cells and immune cells, could cause the vessel narrowing and thickening, which are pathological hallmarks of PH, and, as will be discussed, mitochondrial dysfunction plays a major role in the occurrence of these pathologies [7,8].

As mentioned, when we talk about vascular remodeling, it does not simply entail the structural changes that occurred, but also the molecular mechanisms behind them. Of the many mechanistic insights that have been uncovered during the past decades, the role of mitochondria in driving the progression of vascular remodeling has been diligently studied and established as important [9]. Mitochondria are traditionally known as the “powerhouse” of the cells, where they are vital in producing ATP while also controlling the apoptosis process of a cell [3]. Similar to their role in other cells, mitochondria control cellular homeostasis by the way they can sense whether the cells are deprived of their energy source (e.g., glucose, lipids, or oxygen) and adjust their reaction to this. The adjustment is achieved by initiating apoptosis to maintain the overall homeostasis of a specific multicellular component in the body, due to the fact that mitochondria-driven apoptosis is a mechanism to maintain efficient energy source usage [10].

Furthermore, mitochondria are metabolic sensors and, as such, are integral in both life (ATP production) and death (apoptosis) processes [11]. If the fuel demand of proliferative cells exceeds the fuel supply (glucose, lipids, oxygen), apoptosis may be initiated to preserve fuel or perhaps prevent the transfer of damaged DNA from the oxidative stress of a malfunctioning cell to daughter cells [12]. Mitochondria-dependent apoptosis maintains metabolic fuel efficiency and homeostasis in a multicellular organism, and its suppression may offer a survival advantage to proliferating cells [12]. Another important role of the mitochondria, which is linked with energy source deprivation, is the capability of mitochondria to “switch” energy production in accordance with the availability of its source [13]. This metabolic switch from the regular oxygen-dependent aerobic pathway to the oxygen-independent anaerobic pathway occurs via glycolysis. The glycolytic pathway converts the glucose available in the cell into pyruvate, a process that produces a smaller amount of ATP in comparison to the regular aerobic pathway [14]. On the other hand, the regular aerobic pathway of ATP production continues after the initial glycolysis pathway, where the produced pyruvate is decarboxylated into acetyl-CoA and enters the Krebs cycle [15]. The electrons from NADH and FADH2, produced by the Krebs cycle, flow down the electron transport chain and finally reduce oxygen, causing the secretion of hydrogen ions and energy, which is used to create a larger number of ATPs by ATP synthase [15]. Beyond ATP production, mitochondria are also responsible for a number of other metabolic functions. These include the pentose phosphate pathway, glutaminolysis, fatty acid synthesis and oxidation, and iron metabolism [16].

Additionally, mitochondria have several mechanisms to survive and adapt to the demands and conditions of the cells. These include mitochondria biogenesis, fission, fusion, and mitophagy [17]. First, in mitochondria biogenesis, where the cells increase the number of mitochondria, the synthesis of mitochondrial DNA (mtDNA) occurs [18]. This process is mediated by PGC-1α (peroxisome proliferator-activated receptor gamma coactivator 1-alpha) and TFAM (mitochondrial transcription factor A) genes, where PGC-1α, when activated by external inducers, will stimulate NRFs (nuclear respiratory factors) and subsequently increase TFAM expression [18]. TFAM is the gene that will transcribe and replicate mtDNA, thereby enabling the increase of the mitochondria number in response to the needs of the cell. Sirtuin 1 (SIRT1) is another gene that can modify the biogenesis process by deacetylating and activating PGC-1α, while nitric oxide (NO) has also been reported to promote biogenesis [19,20]. Notably, SMCs have been especially noted as a cell type where changes in biogenesis-related genes play a vital role, especially in modulating their proliferative capability [21]. In contrast to biogenesis, mitophagy is a process in which mitochondria undergo selective autophagy when they are deemed to be irreparably damaged [22]. When this is detected, the outer membrane proteins of the mitochondria will be ubiquitinated and mitophagy will be activated. This process is mediated by several proteins such as Parkin, PINK1 (PTEN-induced kinase 1), and TIM23 (translocase of inner mitochondrial membrane 23) [23].

To further add to the mitochondrial quality control functions, mitochondria are also capable of continuously dividing (termed mitochondrial fission) or fusing (mitochondrial fusion), in accordance with the environmental demands in the cell [17]. While mitochondrial fusion is meant to optimize the work of mitochondria by transferring gene products, mitochondrial fission is aimed at maintaining its proper amount and distribution [17]. Mitochondrial fission depends on several proteins, such as DRP1 (dynamin-related protein 1), FIS1 (mitochondrial fission1), and MFF (mitochondrial fission factors), where DRP1 notably has a central role in its regulation process [24,25]. On the other hand, mitofusins 1 and 2 (MFN1 and MFN2), together with OPA1 (optical atrophy 1), regulate mitochondrial fusion, where MFNs regulate inner membrane fusion and OPA1 regulates outer membrane fusion [26].

In the context of the pulmonary vasculature, different layers of the vasculature have different percentages of mitochondrial content and, accordingly, there are differences in the usage of aerobic and anaerobic respiration between cells. For example, due to the fact that endothelial cells contain fewer numbers of mitochondria in comparison to smooth muscle cells, they relatively rely more on glycolysis in comparison to SMCs [16]. Furthermore, different vascular beds also differ in their source of energy. Whereas the microvasculature ECs rely more on glycolysis, the ECs in the larger pulmonary arteries can depend on aerobic respiration as their source of energy [27]. This is important in allowing the pulmonary vasculature to serve as an oxygen sensor, and the aforementioned capability to switch to glycolysis is also important in mediating the signaling cascade required for hypoxic vasoconstriction of the lung [28]. In the case of pulmonary SMCs, aerobic respiration occupies a major percentage of their energy source, and, as such, they are traditionally the more analyzed cell type when analyzing mitochondrial functions and dysfunctions in a PH setting [29]. Additionally, other types of cells that are important in the maintenance of pulmonary vasculature are also influenced by the work of the mitochondria, such as mononuclear immune cells or adventitial fibroblasts [30,31].

Lastly, mitochondria are also a major source of reactive oxygen species (ROS), so much so that they can essentially regulate the redox status of the cells [4,32]. It is estimated that 90% of ROS generated in the cells originates from mitochondria. During the ETC process in the aerobic respiration process, oxidative phosphorylation, which originally converts O_2_ into H_2_O, will produce leaked electrons at complex I and complex III. The leaked electrons will then interact with molecular oxygen to generate superoxide anions (O_2_^−^), which are emitted into both the matrix and intramembrane space of the mitochondria. Matrix superoxides can be converted into H_2_O_2_ by superoxidases (SOD) 2 and could be reduced into water (H_2_O), while intramembrane superoxides will be converted into H_2_O_2_ by SOD1 [33]. H_2_O_2_, as will be discussed later, is a potent ROS that is reported to be important as a signaling molecule [34]. ROS production in the mitochondria is tightly regulated by several other processes in addition to the SODs, such as ROS scavenging by several enzymes (e.g., catalase, peroxiredoxins, and glutathione peroxidases), alterations in the ETC process, or the electrical and chemical barrier gradient of the inner mitochondrial membrane [4]. This production of ROS could also serve as a trigger for inducing mitochondria-driven apoptosis, as the mitochondrial ROS (mROS) can reach the Kv channels in the plasma membrane, which is redox-sensitive, regulate the voltage-gated Ca2+ channels that govern its influx into the cells, and have an important effect on cellular proliferation and apoptosis [4,34]. As will be explained later, this will be important in the context of vascular remodeling in PH, as both processes are vital parts of said remodeling, and imbalanced mROS production could promote vascular remodeling. 

### 2.2. Mitochondria-Driven Changes in Pulmonary Vascular Remodeling

The importance of mitochondria in maintaining normal cellular function cannot be overstated; its dysfunction could lead to potentially damaging consequences for the cells. Overall, the dysfunction could arise due to changes in many processes in normal mitochondria, such as aberrant ETC protein expression, alterations in the apoptotic process, aberrant mitochondrial dynamics, and excessive ROS production [3,16]. In PH, it is already established that while, on the surface, structural changes are the direct cause of increased pulmonary artery pressure, a complex and interlinked mechanism occurs underneath the visible pathological changes [35]. Mitochondria are one of the most essential organelles to cell homeostasis, and, as such, mitochondrial dysfunction has been linked to many of the mechanisms implicated in pulmonary vascular remodeling. First, we have briefly discussed how a metabolic switch between the aerobic and the anaerobic pathways could occur during specific conditions, such as during low oxygen concentrations. In PH conditions, the metabolic switch does not exclusively occur in said low concentrations, but also in normal oxygen concentrations [13]. This phenomenon, termed the Warburg effect or aerobic glycolysis, was first observed in cancer cells but has also been found in the pulmonary artery ECs (PAEC) and pulmonary artery SMC (PASMC) of patients with idiopathic PAH (IPAH), thus underlying the similarities between PH and cancers [36]. A similar phenomenon could also be found in various animal models of PH, such as those in monocrotaline-induced PH, chronic hypoxia-induced PH, SU5416-Hypoxia rats, and even in fawn-hooded rats [37,38,39].

The occurrence of persistent aerobic glycolysis in PH is thought to be necessary for the continuous growth and production of new cells in cancer, and it is no different in PH conditions, where vascular remodeling is hallmarked by a hyperproliferative, anti-apoptotic phenotype of vascular cells [35,37]. Most notably, the smooth muscle cells of the pulmonary vessels are found to be a major site of this mitochondria-led PH phenotype. Some of the implicated genes in the proliferative and apoptosis-resistant phenotype include survivin, NFAT (nucleic factor of activated T-cells), and hypoxia-inducible factors (HIFs) [3,40]. Furthermore, the downregulation of Kv channels could also be observed in remodeled SMCs [41]. Dysfunction in the ETC process during aerobic respiration is also reported to be able to induce a glycolytic switch in PASMC. For example, one study highlighted how inhibiting mitochondrial respiratory complex III via antimycin A treatment could lead to chronic pulmonary vasoconstriction, a robust increase in lactate and ribose production in the vascular cells that include both SMCs and ECs, and pulmonary hypertension in mice [42]. Fawn-hooded rats, which develop PH, similarly showed dysmorphic mitochondria with reduced complexes I and III component expression. As will be underlined further in this segment, PASMC is one of the main sites of mitochondrial dysfunction, and any imbalance or deficiency in normal mitochondrial functions can adversely impact its phenotype and, subsequently, affect vascular remodeling.

The ECs, on the other hand, have become increasingly established as another important site of mitochondria dysfunction. As mentioned previously, ECs contain a relatively low number of mitochondria in comparison to SMCs and were originally thought to have a negligible metabolic effect, even during mitochondria dysfunction, due to the fact that they mainly rely on anaerobic glycolysis [16,27]. However, recent reports suggest that, rather than having a minimal impact after metabolic reprogramming, major alterations occurred in PAECs that affect their transcriptional and functional phenotype [43]. For instance, various animal models of PH have shown that, similar to PASMCs, PAECs also undergo similar metabolic reprogramming towards aerobic glycolysis. This is thought to occur as a tolerance mechanism towards the chronic hypoxia state, which the cells are under during the PH condition [44]. Several of the important mechanisms include the stabilization of HIFs 1 and 2, apoptosis suppression by hyperpolarized mitochondrial membrane potential, the aforementioned ETC dysfunction in complexes I and III, and NFAT induction with a subsequent increase in glycolytic pathway activation [16,45]. All of these contribute to the uncontrolled proliferation of apoptosis-resistant ECs and endothelial-to-mesenchymal transition found in the remodeled vessels.

Not only does it lead to becoming a less efficient energy producer, the process of aerobic glycolysis also leads to a sudden increase in reactive oxygen species production by the mitochondria. The effect of prolonged oxidative stress in vascular cells is already well-known to promote remodeling through its capability of modulating various functions of particular cells [46]. Oxidative stress has been extensively studied as a phenomenon that is highly related to mitochondria dysfunction and has been discussed elsewhere [47]. Although produced in relatively small amounts, even in normal conditions, an imbalance in mitochondrial ROS has been strongly correlated with PH development in various conditions. Indeed, the majority of PH animal models have confirmed the presence of oxidative stress in the pulmonary vasculature that is concurrent with the hemodynamic and vascular remodeling changes in respective models [47,48,49,50]. In the case of mitochondrial ROS, H_2_O_2_ has been a major focus of study, due to its established role in inflammation, apoptosis, and overall signaling of the cells [51]. Although by simple logic, an increase in mROS production should be seen in vascular remodeling, a line of thinking that has been proven by various studies in ECs and SMCs is that, as some reports suggest, mROS is actually reduced, rather than increased, in the vascular cells, notably in the PASMCs [33,52]. Indeed, a decreased mROS level can be seen in SOD2 deficiency, which leads to the stabilization of HIF-1α due to reduced redox capability [53,54]. Another cause of reduced mROS formation is the dysfunction in ETC that disrupts the aerobic respiration process. Further complicating this is the fact that increased mROS level could also lead to HIF-1α stabilization, as seen in the FUNDC1 (FUN14 domain containing 1)-overexpressing PASMCs [55]. As such, mROS remains a controversial topic to discuss, and more insight is needed to truly uncover its role in vascular remodeling.

Beyond the persistent metabolic switch of the Warburg effect, other metabolic changes also occur as a consequence of mitochondrial dysfunction. For instance, defects in mitochondria could lead to an augmented pentose phosphate pathway (PPP) in parallel with augmented glycolysis [16]. The increase in PPP activity can be found in various animal models of PH and is thought to be a response against increased ROS, as well as being found in hyperproliferative PASMCs [56]. Glutaminolytic reprogramming, in which increased levels of glutamate can be observed, is increasingly common in dysfunctional vascular cells [57]. Iron deficiency is another feature that has been keenly observed in PH models, while findings regarding the role of NFU1 dysfunction related to mitochondrial dysfunction and pulmonary hypertension have been revealed elegantly in several studies. A study by Niihori et al. found that human NFU1 mutation in rats could lead to pulmonary hypertension due to defects in iron-sulfur metabolism [58]. Lastly, alterations in fatty acid metabolism have been observed in both PAEC and PASMCs of PH models [59,60]. Especially in ECs, diminished fatty acid oxidation seems to lead to increased endothelial-to-mesenchymal transition, a hallmark of vascular remodeling [60].

Furthermore, the dynamics of mitochondria through its biogenesis, mitophagy, fission, and fusion could also be impaired and affect vascular remodeling [17]. For example, an imbalance in the fission and fusion of mitochondria could lead to a reduced number of mitochondria due to excessive fission, which can lead to excessive mitophagy [61]. This is proven by reports that linked DRP1 to not only increased mitochondrial fission but also increased mitophagy in PH, among other effects. Another example of imbalanced mitochondrial dynamics happens when overactivation of the mitochondrial fusion pathway occurs and leads to reduced PASMC proliferation and increased apoptosis. This phenomenon occurs in mice with MFN2 overexpression, while in MFN2-deleted mice, increased proliferation of PASMC occurs [62]. Similarly, MFN1 is also linked with an excessive proliferation of PASMCs [63]. The PASMCs are also the site where excessive mitophagy can occur; such is the case in the FUNDC1-overexpressed PASMCs mentioned above [55]. Interestingly, there is a link between each process, as already highlighted by the link between mitochondrial fission and mitophagy, and between fission and fusion. Furthermore, molecules important for biogenesis, such as PGC-1α, have also been linked to MFN2, implying that the process of mitochondrial regulation within the cell is a complex and intertwined system whose delicate balance must be protected to avoid pathological consequences, such as vascular remodeling, occurring [62].

Of course, mitochondrial dysfunction does not solely occur in the remodeled ECs or SMCs. It could also occur in other cells that play a role in maintaining vascular homeostasis. For example, mononuclear immune cells, such as recruited macrophages, also undergo metabolic changes and actively participate in vascular remodeling in PH [64]. An accumulation of macrophages has been observed in various PH animal models, and it has been reported that this accumulation is required for vascular remodeling [65,66]. The metabolic reprogramming that occurs in macrophages is related to the pro-inflammatory activation of said macrophages, where a shift towards aerobic glycolysis occurs, and a reprogramming of the Krebs cycle to increase succinate levels increases ROS formation, and, in the end, promotes pro-inflammatory cytokine generation [67]. HIF-1 stabilization would also occur due to the increase of succinate production, and this would contribute to the expressional increase of pro-inflammatory cytokines of macrophages, thereby “activating” the macrophage into a phenotype that can promote inflammation [68].

Another cellular type that has not been mentioned is the lung mesenchymal cells, such as fibroblasts. As fibroblasts are relatively less metabolically active compared to other cells, they are previously thought to have a minimal role in metabolic changes in PH and in vascular remodeling. However, many reports have subsequently shown how fibroblasts, especially those in the adventitial layer of the vessels, also play vital roles in vascular remodeling [39,69]. During the remodeling process, adventitial fibroblasts have been reported to be alternatively activated, which enables them to be a place for macrophages to attach [70]. Further, fibroblasts themselves could be activated and undergo metabolic reprogramming, contrary to what was previously thought. Increased activation of STAT3 and HIF-1, together with increased expression of IL-6, VEGFA, and GLUT1, points to a shift in metabolic phenotypes in these fibroblasts, several molecules which also contribute to macrophage activation and remodeling in other vascular cell layers [39,70]. In short, mitochondrial dysfunction is an integral process in vascular remodeling with a multitude of consequences at the cellular level. The impact of mitochondrial dysfunction in various cell types of the pulmonary vasculature is illustrated in Figure 1.

## 3. Mitochondrial Dysfunction in Various Forms of Pulmonary Hypertension

The clinical evidence of mitochondrial dysfunction being involved in PH has been known for some time, but as the basic and translational science about the consequences of mitochondrial dysfunction is elucidated more deeply, the more clues appear of how intertwined the pathophysiology of PH actually is among etiologies. Mitochondrial changes are visible in various cellular components that maintain pulmonary vascular homeostasis, including Ecs, SMCs, mesenchymal cells, and even immune cells [35]. For instance, we have mentioned how proof of Warburg effect occurrence could be found in the PAECs and PASMCs of patients with IPAH [36]. The two cells exhibited an increased aerobic glycolysis pathway, evidenced by the increase in glycolysis-related proteins, such as an increase HIF-1α and HIF-2α stabilization, or the subsequent increase in pyruvate dehydrogenase kinase (PDK) [13,16,36]. Even in adventitial fibroblasts of PH patients, evidence of mitochondrial dysfunction could be seen, such as the aforementioned increase in STAT3 and HIF-1 [71].

Not only the Warburg effect, but several other consequences of mitochondria dysfunction could also be observed in PH patients. For example, PH patients showed defective mitochondrial biogenesis, and, subsequently, proper production of mtDNA is also defective [72]. Again, in hyperproliferative lesions found in PAH patients, evidence of mitochondrial damage could be seen through upregulation in ROS, among other effects. In the context of mROS, there are ample clinical findings that support the presence of oxidative stress through the increased level of various markers [73]. These include ADMA (Asymmetric dimethylarginine) and 8-OhDg, among others [74]. Another way to analyze mROS overactivation is to analyze the expression levels of ARE (antioxidant response element), a collection of genes that could be upregulated after oxidative stress and subsequent translocation of NRF-2 to the nucleus [75]. As mentioned, mROS could also act as a signaling molecule, and overproduction of mROS could lead to aberrant signaling. One example of signaling that could be triggered by mROS is inflammatory signaling via NFκB (nuclear factor kappa B) signaling. Indeed, numerous reports have linked an overabundance of H_2_O_2_ to triggering NFκB-driven inflammatory responses. Notably, cytokine expression is triggered after NFκB, which includes several cytokines already established as important in PH pathogenesis such as TNF-α, IL-1β, and IL-6. TNF-α has been reported as an important driver for BMPRII transcriptional reduction in PASMC, in addition to its pro-inflammatory property, whereas overexpression of IL-6 is known to induce PH [76,77]. Furthermore, some of the cytokines, for example, TNF-α, are also able to induce further mROS production from further mitochondrial damage, establishing a dangerous cycle that leads to the remodeling of the vascular cells [78]. Inflammation is an important and established mechanism underlying various forms of PH, as is the upregulation of pro-inflammatory cytokines. Beyond ROS, other metabolic changes in the vascular cells of PH patients could be observed. This included, for instance, increased glutamate levels in the PASMCs of PAH patients as a result of aberrant glutamine metabolism, and a tendency for PH patients to have an iron deficiency [79,80]. The G208C mutation in the NFU1 gene has been reported to reduce the activity of respiratory complex II in the mitochondria, while roughly 70% of its carriers develop PAH. Further, many studies have indicated that an increase in FA synthase, which converts malonyl-CoA to free fatty acids, is increased in the PAECs and PASMCs of PAH patients [16,60].

Notably, as mitochondria are a chief regulator for various metabolic processes, it is now thought that the metabolic phenotype that could be found in PH patients is also related to mitochondrial dysfunction. Indeed, recently several metabolic syndrome phenotypes could be observed in PH patients, such as dyslipidemia, obesity, or insulin resistance [81]. This phenotype could be found in different groups of PH, not only limited to the obvious group 2 PH, due to left heart disease, but also in group 3 PH, due to chronic hypoxia/lung disease, or even in group 1 PAH. Even more notable is how several of the known genes important in metabolic syndrome pathophysiology, such as UCP2, PPARG, or SIRT3, are also associated with the PH condition clinically, which is corroborated by pre-clinical study results [21,81,82]. Mitochondrial dysfunction has long been related to metabolic syndrome, and the findings of metabolic syndrome in PH reinforce the evolving belief that PH is not isolated to the pulmonary vasculature, but quite possibly has systemic implications. As such, analyzing the mitochondrial and metabolic dysfunctions in PH patients of varying etiologies is important for future clinical considerations, including its classification and treatment strategies.

## 4. Targeting Mitochondrial Dysfunction to Treat PH

As underlined in the previous sections, the vast changes in mitochondrial functions have been a major source of problems in PH conditions, and thus targeting this particular phenomenon could be an important part of PH therapies. Notably, various therapeutic agents targeting mitochondria have been tested in PH patients after successful preclinical studies, unfortunately with mixed results so far. An example of this is dichloroacetate (DCA) [83]. DCA is aimed at inhibiting PDK2 and upregulating PDH activity, thereby redirecting the energy production pathway back to the aerobic respiration pathway rather than glycolysis [84]. Although this treatment is effective in animal PH models, consistent results could not be observed in PAH patients. Another example of a mitochondria-targeting agent is ranolazine, an FAO (fatty acid oxidation) inhibitor [85]. Targeting FAO is thought to be able to improve PH condition via indirect glucose oxidation promotion through the aforementioned DCA activation and PDH activity upregulation. This line of thought was proven in MCD (malonyl-CoA decarboxylase)-deficient mice that were resistant to PH, even after chronic hypoxia. A clinical study showed how ranolazine could improve clinical symptoms and RV functions of PAH patients without any significant hemodynamic improvements, results that, at a glance seem contradictory [85]. Another agent targeting FAO is trimetazidine, a drug that is already approved as an anti-angina agent. In vivo results in chronic hypoxia and monocrotaline-induced PH animal models showed promising results, while the clinical evidence for its use is still under trial [86,87]. NFAT inhibitors, as they are important regulators of glycolysis, have also been thought of as a therapeutic agent for PAH. Cyclosporine, for instance, has been reported to be able to reverse PAH in rat models [88,89]. Since cyclosporine has been used as an immunosuppressant in various conditions, such as psoriasis and rheumatoid arthritis, it would be interesting to see whether similar efficacy could be translated to human patients in the future.

Beyond targeting the metabolic changes induced by mitochondria dysfunction and subsequent occurrence of the Warburg effect, other defective processes involving the mitochondria, such as mitochondrial dynamics and biogenesis, have also been tried as therapeutic targets in pre-clinical settings, although clinical results have been scarce as well. For example, targeting mitochondrial biogenesis through the activation of NRF-1 and HO-1 was successful in experimental animal models, similar to activating the CO/HO-1 pathway [90]. Another pathway that is promising as a mitochondrial biogenesis inducer is the AMPK pathway, where its activation could lead to increased biogenesis. Several AMPK activators have been established in pre-clinical settings to positively regulate mitochondrial functions and dynamics [91]. Although these results hold promise, clinical evidence is still needed to be sure of its efficacy in treating PH.

Lastly, targeting oxidative stress induced by mitochondrial dysfunction has also been tried in both pre-clinical and clinical settings. In animal models of PH, several ROS-targeting treatments have been successful in reversing PH phenotypes. An example of this is MitoQ, which targets mitochondrial ROS [92]. Although, in chronic hypoxia, it did not cause amelioration of PH, in acute hypoxia, MitoQ treatment successfully prevented hypoxic pulmonary vasoconstriction, a physiological event that is persistently found in PH. MitoTEMPO is another agent targeting mROS, in this case by being a SOD2 mimetic [93]. Unfortunately, although many successful and effective treatments to ameliorate oxidative stress have been found in animal studies, such as those found in fawn-hooded rats, similar encouraging results have not been found in humans. Other ROS-targeting treatments, such as SS31 or NRF2-activating compounds, have also been tried on a pre-clinical basis but have yet to be applied in patients [3]. We have summarized the current therapeutic options targeting mitochondrial dysfunctions in Figure 2. In short, although many pathways related to mitochondrial dysfunction are available to target in PH, clinical evidence is still lacking and, in this aspect, more studies are sorely needed.

## 5. Conclusions

There is no denying that mitochondria serve as important regulators of vascular cell homeostasis, and the failure of normal mitochondrial work leads to vascular remodeling and pulmonary hypertension. Future studies are needed to fully realize the potential of intervening in mitochondrial dysfunction for treating PH.

## Figures and Tables

**Figure 1 antioxidants-12-00372-f001:**
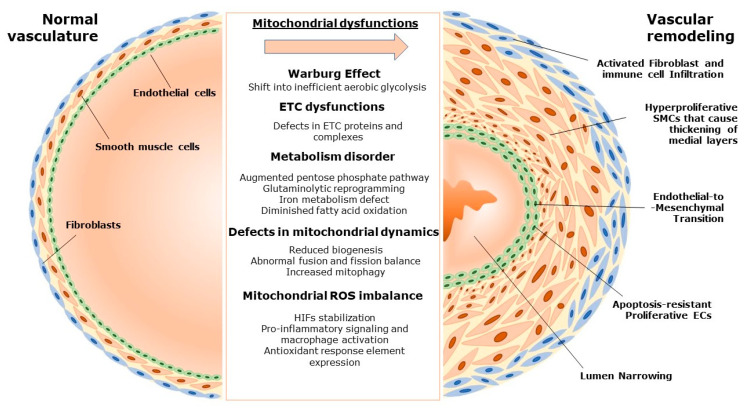
Schematic summary of mitochondrial dysfunctions and their effects on pulmonary vascular remodeling. ETC, electron transport chain; SMCs, smooth muscle cells; ROS, reactive oxygen species; ECs, endothelial cells; HIF, hypoxia-inducible factor.

**Figure 2 antioxidants-12-00372-f002:**
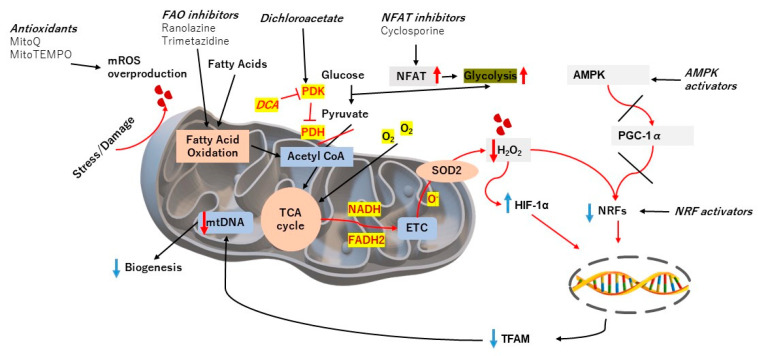
Current therapeutic agents targeting mitochondrial dysfunction in PH conditions. FAO, fatty acid oxidation; NFAT, nuclear factor of activated T-cells; AMPK, AMP-activated protein kinase; PGC-1α, peroxisome proliferator-activated receptor gamma coactivator 1-alpha; NRF, nuclear factor erythroid 2-related factor; HIF, hypoxia-inducible factor; SOD, superoxide dismutase; mROS, mitochondrial reactive oxygen species; ETC, electron transport chain; TFAM, transcription factor A, mitochondrial.

## Data Availability

Data are contained within this article.

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
