# Peer review of "Mitochondrial Dysfunction in Pulmonary Hypertension"

_antioxidants, 2023, doi:10.3390/antiox12020372_

Round 1

Reviewer 1 Report

The overall impression of the contribution of the current study “Mitochondrial Dysfunction in Pulmonary Hypertension” is reasonable. However, the Authors may consider doing necessary amendments to the manuscript for better comprehensibility of the study.

I would suggest authors to include figures summarizing different sections, for example for Mitochondria and vascular homeostasis and Mitochondria-driven changes in pulmonary vascular remodeling sections. Now, there is none.

Please discuss the role of cytokines (TNFα), ETC dysfunction, NFU1 gene mutations in this context.

Please provide a schematic presentation summarizing the mechanism and, if possible on potential role of antioxidants for Mitochondrial Dysfunction in Pulmonary Hypertension

Please check carefully for typos.

For example please change the title “Mitochondrial Dysfunction in Pulmonary Hypertension:”, to “Mitochondrial Dysfunction in Pulmonary Hypertension”

Author Response

Response to reviewers

We thank the reviewers and editor for their comments and suggestions for our manuscript. We feel that with the input that we receive, we are able to improve the quality of our manuscript greatly. Below is our point-by-point response to the reviewers’ comments.

Reviewer 1

The overall impression of the contribution of the current study “Mitochondrial Dysfunction in Pulmonary Hypertension” is reasonable. However, the Authors may consider doing necessary amendments to the manuscript for better comprehensibility of the study.

Response: Thank you very much for your encouraging comments. We are grateful for the constructive comments that you have provided for our manuscript.

I would suggest authors to include figures summarizing different sections, for example for Mitochondria and vascular homeostasis and Mitochondria-driven changes in pulmonary vascular remodeling sections. Now, there is none.

Response: Thank you for your important suggestion. We have added the figure that summarize the mitochondria dysfunctions and the vascular changes it causes in figure 1 in page 7, while additionally showing illustration of the various molecular pathways related to mitochondrial dysfunction in figure 2 in page 9.

Please discuss the role of cytokines (TNFα), ETC dysfunction, NFU1 gene mutations in this context.

Response: Thank you for your suggestions. We have included the discussion about cytokines in page 8 line 327-336, ETC dysfunctions in page 4 line 183-190, page 5 line 206-209 and page 5 line 227-230, and NFU1 gene mutations in page 5 line 240-244 and page 8 line 341-343.

Please provide a schematic presentation summarizing the mechanism and, if possible on potential role of antioxidants for Mitochondrial Dysfunction in Pulmonary Hypertension

Response: Thank you for your comments. We have inserted a schematic illustration of the various molecular pathways related to mitochondrial dysfunction and the placement of therapeutic agents targeting specific pathways, including antioxidants, in figure 2 on page 9.

Please check carefully for typos.

For example please change the title “Mitochondrial Dysfunction in Pulmonary Hypertension:”, to “Mitochondrial Dysfunction in Pulmonary Hypertension”

Response: Thank you for your comments. We have corrected the typo in the title and also throughout the manuscript.

Reviewer 2 Report

In this review manuscript, the authors discuss the role of mitochondrial dysfunctions in the development of PH. Specifically, clinical evidence of mitochondrial dysfunctions, multiple involved molecular signaling pathways, and several therapeutic agents targeting mitochondria in clinical trials have been fairly meticulously discussed. However, the current version of the manuscript have a number of questions and comments that the authors need to address thoroughly, as described below.

1.     There are numerous typos, grammatical errors, misspellings, convoluted sentences, and sentence structure existed throughout the manuscript. Some examples are given below.

Page 1 of 13

Line 13: Add “The” and capitalize Effect – “The Warburg Effect…”

Line 40: Capitalize the World Health Organization (WHO)

Page 2 of 13

Line 50: Add “s” after EC (ECs) and SMC (SMCs), indicating plural cells

Line 53: Please correct to either “this pathology” or “these pathologies”

Line 60: Change “on” to “in”.. “Similar to its role in other cells”, and change “controls” to “control”

- You do not need to say “one way”, “in other cells, mitochondria control cellular homeostasis by sensing whether cells are deprived of its energy…”

- Sentence is very long, consider breaking up into 2 sentences

Line 66: The word “Mitochondria” does not need to be capitalized

Line 77: Use “oxygen-independent anaerobic…”

Line 91: Change “could be” to “is”

Line 95: Change to “deacetylating and activating PGC-1a” – ending sentence with “it” is ambiguous

Line 96: Move “also” after “has” – “(NO) has also been reported…”

Line 97: Not sure I would consider SMCs a “place” – try “cell type”

Page 3 of 13

Line 99: Elementary style writing – change to “In contrast to biogenesis, mitophagy is a process in which mitochondria undergo…”

Line 102: Capitalize Parkin

Line 104: Reword beginning – “To further add to mitochondrial quality control functions, ...”

Line 131 and 146: Used “so much so” too close together, try another phrase

Page 4 of 13

Line 153: Change “dysfunction to the mitochondria” to “mitochondrial dysfunction”

Line 155: Change to “metabolic switching” or “how a metabolic switch between…”

Line 157: Change to “conditions” – “In PH conditions..”, edit “the metabolic switch does not exclusively occur in low conditions”

Line 172: Insert “channels” after KV, insert “it” after as – “As it will be underlined further…”

Line 180: Change “suggests” to “suggest” – “However, recent reports suggest that …”

Line 194 and 195: Uses the word “extensively” twice in the same sentence

……

Page 7 of 13

Line 326: Change “line of thinking” to “line of thought”

Line 327: Remove “the” before “MCD-deficient mice”

Line 341: Change to “processes” – “other defective processes involving…”

Page 8 of 13 – Conclusions

Conclusions paragraph is relatively short for the length of the paper, authors may consider adding a schematic diagram highlighting the important pathways and function of mitochondria as thoroughly discussed in the review article.

2.     The abbreviations such as PGC-1α, TFAM and many others should only be used if they appear two or more times in the text; moreover, the full name should be given at its first mention, and the abbreviation is used from then on.

3.     Many signaling molecules are included; however, some of them have not been properly discussed. For instance, How do mitochondria produce ROS, and how important are mitochondrial ROS?

4.     A number of upstream and downstream signaling pathways are not very clearly described; nevertheless, a few diagrams should be added for clarification.

Author Response

Response to reviewers

We thank the reviewers and editor for their comments and suggestions for our manuscript. We feel that with the input that we receive, we are able to improve the quality of our manuscript greatly. Below is our point-by-point response to the reviewers’ comments.

Reviewer 2

In this review manuscript, the authors discuss the role of mitochondrial dysfunctions in the development of PH. Specifically, clinical evidence of mitochondrial dysfunctions, multiple involved molecular signaling pathways, and several therapeutic agents targeting mitochondria in clinical trials have been fairly meticulously discussed. However, the current version of the manuscript have a number of questions and comments that the authors need to address thoroughly, as described below.

Response: Thank you very much for your kind words and comments. We are thankful for the comments that you have provided for our manuscript.

  1. There are numerous typos, grammatical errors, misspellings, convoluted sentences, and sentence structure existed throughout the manuscript. Some examples are given below.

Page 1 of 13

Line 13: Add “The” and capitalize Effect – “The Warburg Effect…”

Line 40: Capitalize the World Health Organization (WHO)

Page 2 of 13

Line 50: Add “s” after EC (ECs) and SMC (SMCs), indicating plural cells

Line 53: Please correct to either “this pathology” or “these pathologies”

Line 60: Change “on” to “in”.. “Similar to its role in other cells”, and change “controls” to “control”

- You do not need to say “one way”, “in other cells, mitochondria control cellular homeostasis by sensing whether cells are deprived of its energy…”

- Sentence is very long, consider breaking up into 2 sentences

Line 66: The word “Mitochondria” does not need to be capitalized

Line 77: Use “oxygen-independent anaerobic…”

Line 91: Change “could be” to “is”

Line 95: Change to “deacetylating and activating PGC-1a” – ending sentence with “it” is ambiguous

Line 96: Move “also” after “has” – “(NO) has also been reported…”

Line 97: Not sure I would consider SMCs a “place” – try “cell type”

Page 3 of 13

Line 99: Elementary style writing – change to “In contrast to biogenesis, mitophagy is a process in which mitochondria undergo…”

Line 102: Capitalize Parkin

Line 104: Reword beginning – “To further add to mitochondrial quality control functions, ...”

Line 131 and 146: Used “so much so” too close together, try another phrase

Page 4 of 13

Line 153: Change “dysfunction to the mitochondria” to “mitochondrial dysfunction”

Line 155: Change to “metabolic switching” or “how a metabolic switch between…”

Line 157: Change to “conditions” – “In PH conditions..”, edit “the metabolic switch does not exclusively occur in low conditions”

Line 172: Insert “channels” after KV, insert “it” after as – “As it will be underlined further…”

Line 180: Change “suggests” to “suggest” – “However, recent reports suggest that …”

Line 194 and 195: Uses the word “extensively” twice in the same sentence

……

Page 7 of 13

Line 326: Change “line of thinking” to “line of thought”

Line 327: Remove “the” before “MCD-deficient mice”

Line 341: Change to “processes” – “other defective processes involving…”

Page 8 of 13 – Conclusions

Conclusions paragraph is relatively short for the length of the paper, authors may consider adding a schematic diagram highlighting the important pathways and function of mitochondria as thoroughly discussed in the review article.

Response: Thank you very much for your detailed and important comments. We have corrected all the typos you have mentioned and reconstructed the sentences as suggested, while additionally inserting schematic diagrams highlighting the pathways involved in mitochondrial dysfunction in figures 1 (page 7) and 2 (page 9).

  1. The abbreviations such as PGC-1α, TFAM and many others should only be used if they appear two or more times in the text; moreover, the full name should be given at its first mention, and the abbreviation is used from then on.

Response: Thank you very much for your comments. We have corrected the abbreviations and made sure it is only used when it appears more than one time, while full name has been added to each abbreviation at the first mention.

  1. Many signaling molecules are included; however, some of them have not been properly discussed. For instance, How do mitochondria produce ROS, and how important are mitochondrial ROS?

Response: Thank you very much for your comments. We have expanded the explanation of mitochondrial ROS and its importance on pages 3 line 133- 146 and pages 7-8 line 325-338.

  1. A number of upstream and downstream signaling pathways are not very clearly described; nevertheless, a few diagrams should be added for clarification.

Response: Thank you very much for your suggestion. We have added schematic figures 1 and 2 to illustrate the pathways included in mitochondrial dysfunction in pages 7 and 9.

Round 2

Reviewer 1 Report

The authors have addressed all my queries in the revised manuscript, and I don't have any other suggestions. 

Reviewer 2 Report

Well revised!